# COVID-19 hospitalizations in Brazil's Unified Health System (SUS)

**Carla Lourenço Tavares de Andrade, Claudia Cristina de Aguiar Pereira, Mônica Martins, Sheyla Maria Lemos Lima, Margareth Crisóstomo Portela** *

Departamento de Administração e Planejamento em Saúde, Escola Nacional de Saúde Pública Sergio Arouca, Fundação Oswaldo Cruz, Rio de Janeiro, RJ, Brazil

* mportela@ensp.fiocruz.br

**Data Availability Statement:** All relevant data are within the manuscript.

**Funding:** The author(s) received no specific funding for this work.

## Abstract

### Objective

To study the profile of hospitalizations due to COVID-19 in the Unified Health System (SUS) in Brazil and to identify factors associated with in-hospital mortality related to the disease.

### Methods

Cross-sectional study, based on secondary data on COVID-19 hospitalizations that occurred in the SUS between late February through June. Patients aged 18 years or older with primary or secondary diagnoses indicative of COVID-19 were included. Bivariate analyses were performed and generalized linear mixed models (GLMM) were estimated with random effects intercept. The modeling followed three steps, including: attributes of the patients; elements of the care process; and characteristics of the hospital and place of hospitalization.

### Results

89,405 hospitalizations were observed, of which 24.4% resulted in death. COVID-19 patients hospitalized in the SUS were predominantly male (56.5%) with a mean age of 58.9 years. The length of stay ranged from less than 24 hours to 114 days, with a mean of 6.9 (±6.5) days. Of the total number of hospitalizations, 22.6% reported ICU use. The odds on in-hospital death were 16.8% higher among men than among women and increased with age. Black individuals had a higher likelihood of death. The behavior of the Charlson and Elixhauser indices was consistent with the hypothesis of a higher risk of death among patients with comorbidities, and obesity had an independent effect on increasing this risk. Some states, such as Amazonas and Rio de Janeiro, had a higher risk of in-hospital death from COVID-19. The odds on in-hospital death were 72.1% higher in municipalities with at least 100,000 inhabitants, though being hospitalized in the municipality of residence was a protective factor.

### Conclusion

There was broad variation in COVID-19 in-hospital mortality in the SUS, associated with demographic and clinical factors, social inequality, and differences in the structure of services and quality of health care.

**Competing interests:** The authors have declared that no competing interests exist.

## Introduction

The Covid-19 pandemic caused by the SARS-CoV-2 virus has severely affected Brazil, which has become the country with the second highest number of cases and deaths in the world [1]. The first confirmed case of COVID-19 in Brazil and Latin America occurred on February 26, 2020 in the state of São Paulo. Less than a month later, the first death also occurred in São Paulo on March 17. Social distancing measures were first introduced in March in five states, namely Goiás, Rio de Janeiro, Santa Catarina, the Federal District and São Paulo [2]. Due to the rapid spread of the disease, all 26 states and the Federal District had already registered ten or more cases of the disease in early April, with a higher concentration of cases in Southeastern Brazil, especially in the states of São Paulo and Rio de Janeiro [3].

Given the rapid transmission of the virus, there was an abrupt and growing additional demand for hospitalizations worldwide, thus putting health care systems under strain in many countries [4]. According to the World Health Organization (WHO), 80% of patients with COVID-19 have mild and uncomplicated symptoms, 15% progress to hospitalization and 5% require admission to the intensive care unit (ICU) [5].

Brazil's Unified Health System (SUS) is the largest public and universal health system in the world, encompassing the entire country. About 75% of the Brazilian population does not have private health insurance and is exclusively dependent on the SUS [6, 7]. The system's under-funding and inadequate management, however, has undermined its structure, with broad variation in the quality of services provided across the country. The need to cope with COVID-19 has revealed weaknesses in the system, despite the increase in the number of general and intensive care hospital beds on offer and the construction of field hospitals. Several Brazilian states had to deal, to a greater or lesser extent, with a higher demand than the available capacity of the SUS to respond, which resulted, for example, in long queues for general and intensive care beds.

In addition to the structure of the services supplied for COVID-19 cases and the measures implemented to control the pandemic, patient characteristics, such as age, sex, socioeconomic status and pre-existing conditions, interfere with the demand for hospitalization, the care provided and the outcomes [8, 9]. Worldwide, ICU mortality due to COVID-19 in hospitalizations of patients aged 18 years and older is higher than that normally seen in patients with other viral pneumonias [4]. As the pandemic progressed, reported death rates dropped from more than 50% to close to 40%. Comorbidities frequently referred in studies include obesity, hypertension, diabetes mellitus, cardiovascular, lung, chronic kidney and liver diseases, immunosuppression and cancers [4].

In Brazil, a cross-sectional observational study conducted with hospitalized COVID-19 patients identified a lower likelihood of death in young, female patients with fewer comorbidities, commensurable with what has been observed in other countries. Furthermore, it highlighted a higher risk of death among black and mixed race populations and in patients hospitalized in the Northern region compared to other regions of the country, which, for the authors, represents a specific manifestation of the disease in the Brazilian population. The authors suggest that the regional effect may be driven by the morbidity profile of patients in regions with lower levels of socioeconomic development [10].

Despite the acknowledged increase in international and national publications on COVID-19, there are still few studies that examine the risk factors and characteristics of hospitalizations for COVID-19 in the population considering geographic variations [11]. Therefore, the following question arises: How does the profile of patients and hospitalizations observed in the international literature compare with hospitalizations that occurred in Brazil's public health system in different regions of the country? The scope of this paper is to understand the profile of COVID-19 hospital admissions in the Unified Health System (SUS) and to identify associated

factors with the occurrence of in-hospital deaths related to the disease, considering patient characteristics and the care offered, with a focus on regional variations.

## Methods

### Study design

This is a cross-sectional, observational study based on secondary data on COVID-19 hospitalizations that occurred in the Unified Health System (SUS), which were available on the DATA-SUS website on August 4, 2020 [12], considering the first four months of the pandemic in Brazil, namely between the end of February and the last week in June.

The SUS Hospital Information System (SIH) was the data source. Although this system may raise some coverage and quality concerns, as administrative data often do, it is the main source of information on hospital production nationwide and has been employed in other scientific studies. During a pandemic in which evidence still needs to be acquired, and Brazil has unquestionable importance in terms of the number of cases and deaths, the role it can play is not negligible. There is no other data source able to provide the information it can offer within a relatively short timeframe. In addition to demographic data (age, sex), it includes diagnostic data, type of admission (elective/emergency) and type of care (surgical/clinical), length of stay (LOS), use of intensive care (ICU), outcome at discharge and the amount reimbursed for the hospitalization. In 2016, it was expanded to accept up to nine secondary diagnosis registrations, potentially providing a better picture of the morbidity and severity case profile. In addition to secondary diagnoses, variables indicating their pre-existence (presence on admission) or that they were acquired during the process of care in the hospital designate, respectively, that they are patient attributes or results of performance and quality of care problems. Race/color is the only variable available in the SIH dataset that is responsive to socioeconomic conditions.

The data used were extracted from the reduced type (RD) files for each state and Federal District, freely available on the DATASUS portal [12]. It is likely that hospitalizations in the months under scrutiny are underreported due to the specificities of the SIH data transmission process and its main purpose, which is reimbursement.

### Study sample

Initially, we excluded hospitalizations of patients under 18 years of age. The selection of COVID-19 hospitalizations began with the following variables: procedure performed (which designates the type of treatment performed on the patient, be it clinical or surgical, and serves as the basis for hospitalization payment); primary and secondary diagnoses. We considered the patients whose hospital record indicated an association with COVID-19 in any of these variables as cases under scrutiny. Thus, all hospitalizations with the primary diagnosis or one of the secondary diagnoses identified with the International Statistical Classification of Diseases and Related Health Problems, 10th revision (ICD-10) code B34.2 –coronavirus infection of unspecified location–were included. This diagnostic category was defined in the technical guidelines of the SIH in the context of the pandemic [13]. In line with these guidelines, we also included hospitalizations coded "03.03.01.022–3—TREATMENT OF INFECTION BY THE NEW CORONAVIRUS—COVID 19" in the procedure variable [14]. This code was recently created to take effect from April 14, 2020 onwards. In hospitalizations prior to April, the procedure used was "03.03.01.019–3—TREATMENT OF OTHER DISEASES CAUSED BY VIRUSES (ICD-10: B25-B34)." Given its lack of specificity and relation to a wide range of core diagnoses, it was not considered an additional inclusion criterion beyond the B34.2, except for

a few records containing the B97.2—Coronavirus, as the cause of classified diseases in other chapters as the main diagnosis.

## Data analysis

The study focused on the analysis of patient sociodemographic and clinical characteristics, the care process and contextual variables related to the hospitalization, and their effects on the likelihood of in-hospital death.

Descriptive and bivariate analyses were performed to characterize the population studied and to test the relationships between the independent variables (attributes of the patients, the care process and the hospital, place of residence and geographic location) with the dependent 'in-hospital death' variable. The scope of the information available in the dataset shaped the range of operationalized variables and the scope of the analyses.

To account for the correlation among observations occurring in each hospital, resulting from the process of care and case-mix, we used the generalized linear mixed model (GLMM) with random effects intercept. This model was used to assess the different factors associated (independent variables) with the in-hospital death of COVID-19 patients (response with binary distribution). Thus, the modeling occurred, with the insertion of different blocks of variables, in three stages: (i) patient attributes–variables that express the patient risk profile and social inequality (race/color); (ii) elements of the care process; and (iii) characteristics of the hospital, place of residence and place of hospitalization.

For the first stage, the case severity profile was based on demographic variables (sex and age) and comorbidities. The 'sex' variable is binary as informed in the SIH, and we considered female as the reference category. Age was treated as a categorical variable (18–39, 40–49, 50–59, 60–69, 70–79, 80–89 and $\geq$ 90 years). Comorbidities were contemplated in different ways: (i) calculating the Charlson Comorbidity Index (CCI) [15]; (ii) identifying the presence of comorbidities as proposed by the Elixhauser Comorbidity Index (ECI) for administrative databases [16]; (iii) considering specific comorbidities–obesity, arterial hypertension and diabetes [17]–due to their frequency in the population and their relevance in the COVID-19 literature, although they are already part of the previous comorbidity parameters used. CCI and ECI were chosen because they are widely used measures in models for predicting death [17] and have previously been applied to Brazilian patients [18]; the calculation used the ICD-10 coding algorithm for each clinical condition developed by Quan et al. [19]. Additionally, we considered whether COVID-19 was designated as a primary (reference category) or secondary diagnosis. In the case of the 'race/color' variable, we started out by using the five categories provided by SIH (white, black, mixed race, yellow and indigenous) and, after examination, we chose to consider the categories black, mixed race and others, as reference. Race/color is used as a proxy for social vulnerability and inequalities in socioeconomic conditions, health, access, use and effectiveness of care.

In the second stage, two variables about the care process were added to the previous model: ICU use and LOS. The number of days spent in the intensive care unit (ICU) was transformed into a dichotomous variable (yes/no). The LOS was also categorized considering the distribution of deaths. Therefore, the following categories were established: 1 day or less; 2–7 days; 8–22 days and $\geq$ 23 days; hospitalizations in which the LOS was 0, were considered to be less than 24 hours and included in the first category.

In the third and last stage of the modeling process, we inserted hierarchical level variables, namely the hospital, the place of residence and the geographic location where the hospitalization occurred. At the hospital level, the legal nature was also considered, and they were categorized as municipal, state and federal public hospitals, private for profit and private non-profit

hospitals. For the place of residence, a variable indicating patient displacement to seek care was created, considering whether the municipality of residence and the location of the hospital were the same (dichotomous variable yes/no). Lastly, we used categories for each state (including the Federal District) to account for geographic effects in the occurrence of hospitalization; and municipal population size, which, in previous analysis, was more adequate than the municipal human development index (Municipal HDI).

The predictive capacity of the different models was assessed based on the "c" statistic, and data analysis was performed using the SAS statistical package.

## Results

The selection criteria for hospitalizations yielded 89,405 records, among which 13 corresponded to hospitalizations that started and ended in 2020 before the month of March. Altogether, 21,807 (24.4%) hospitalizations resulted in death.

The COVID-19 patients hospitalized in the SUS were predominantly male (56.5%), aged between 18 and 114 years old, mean and standard deviation of 58.9 (± 16.8) and median of 60 years. The LOS ranged from 24 hours or less to 114 days, with an average of 6.9 (± 6.5) days and a median of 5 days. Altogether, hospitalizations represented R$ 332.10 million (Brazilian reais) in expenditures for the SUS, which varied between 40.40 and 111,914.50 reais per patient, with an average of R$ 3,714.50 (± 6,119.20) and first, second and third quartiles were 1,500.00, 1,610.40 and 2,087.20 Brazilian reais, respectively.

Of the total number of hospitalizations, 22.6% registered ICU use, which accounted for 66.4% of the total amount paid for all COVID-19 hospitalizations. In these admissions, the average and median hospital length of stay was 10.3 (± 8.5) days and 8 days, respectively, and the amount paid for average and median hospital stay was 11,083.10 (± 9,674.30) and 8,036.40 reais, respectively. Considering ICU alone, the average LOS was 7.6 (± 6.8) days, with a median of 6 days.

The analyses presented here considered the association between the occurrence of in-hospital death and three "blocks" of variables: sociodemographic and clinical attributes of patients; aspects related to the care process; and data from the macro context of the hospital organization and geographic location of the hospital. Tables 1–3 show descriptive statistics and bivariate analyses of these variables, when death occurred or not.

In Table 1, we observe higher hospital mortality for men and an increasing gradient of the likelihood of dying as age (age group) increases. Among individuals aged 18 to 39 years, 8.5% of hospitalizations resulted in death. For individuals between 70 and 79, 80 and 89, and 90 years old and above, the proportion of hospitalizations that resulted in death increased to 36.0%, 44.5% and 50.8%, respectively.

Mixed race individuals are the majority in the 'race/color' variable, but the high percentage of hospitalizations with unspecified 'race/color' data (28.9%) draws special attention. Among the cases with complete information on the variable, there is a higher occurrence of deaths among blacks (31.9%), followed by indigenous people (28.9%) and mixed race people (26.1%). The percentage of in-hospital deaths with unspecified 'race/color' data is slightly higher than that observed among whites and lower than that observed among mixed race people. It is also worth noting that only 152 admissions of indigenous individuals were observed, comprising 0.2% of the total.

The clinical variables expose low levels of information about secondary diagnoses, resulting in a significant underreporting of clinical conditions relevant to the patients' prognosis. Approximately 78.3% of hospitalizations did not have any secondary diagnoses. Nevertheless, the results indicate a higher occurrence of deaths among patients with greater severity

**Table 1. Bivariate analyses of sociodemographic and clinical variables and in-hospital mortality.** COVID-19 hospitalizations in the Unified Health System in Brazil (N = 89,405). February to June 2020.

| Variables | N | % | In-hospital death | | | | $\chi^2$ (p-value) |
|---|---|---|---|---|---|---|---|
| | | | Yes | | No | | |
| | | | N | % | N | % | |
| Sex | | | | | | | < 0.0001 |
| Male | 50,520 | 56.5 | 12,867 | 25.5 | 37,653 | 74.5 | |
| Female | 38,885 | 43.5 | 8,940 | 23.0 | 29,945 | 77.0 | |
| Age (years) | | | | | | | < 0.0001 |
| 18–39 | 13,028 | 14.6 | 1,113 | 8.5 | 11,915 | 91.5 | |
| 40–49 | 13,556 | 15.2 | 1,730 | 12.8 | 11,826 | 87.2 | |
| 50–59 | 17,724 | 19.8 | 3,234 | 18.2 | 14,490 | 81.8 | |
| 60–69 | 19,098 | 21.4 | 5,363 | 28.1 | 13,735 | 71.9 | |
| 70–79 | 15,593 | 17.4 | 5,615 | 36.0 | 9,978 | 64.0 | |
| 80–89 | 8,483 | 9.5 | 3,775 | 44.5 | 4,708 | 55.5 | |
| ≥ 90 | 1,923 | 2.1 | 977 | 50.8 | 946 | 49.2 | |
| Ethnic group | | | | | | | < 0.0001 |
| White | 21,260 | 23.8 | 4,635 | 21.8 | 16,625 | 78.2 | |
| Black | 5,507 | 6.2 | 1,754 | 31.9 | 3,753 | 68.1 | |
| Mixed race | 33,542 | 37.5 | 8,741 | 26.1 | 24,801 | 73.9 | |
| Yellow | 3,071 | 3.4 | 595 | 19.4 | 2,476 | 80.6 | |
| Indigenous | 152 | 0.2 | 44 | 28.9 | 108 | 71.1 | |
| Unspecified | 25,873 | 28.9 | 6,038 | 23.3 | 19,835 | 76.7 | |
| Charlson Index | | | | | | | < 0.0001 |
| 0 | 86,131 | 96.3 | 20,560 | 23.9 | 65,571 | 76.1 | |
| 1 | 2,552 | 2.9 | 898 | 35.2 | 1,654 | 64.8 | . |
| ≥ 2 | 722 | 0.8 | 349 | 48.3 | 373 | 51.7 | |
| Elixhauser comorbidities | | | | | | | < 0.0001 |
| Yes | 5,574 | 6.2 | 1,887 | 33.9 | 3,687 | 66.1 | |
| No | 83,831 | 93.8 | 19,920 | 23.8 | 63,911 | 76.2 | |
| Obesity | | | | | | | < 0.0001 |
| Yes | 655 | 0.7 | 224 | 34.2 | 431 | 65.8 | |
| No | 88,750 | 99.3 | 21,583 | 24.3 | 67,167 | 75.7 | |
| Hypertension | | | | | | | < 0.0001 |
| Yes | 3,794 | 4.2 | 1,273 | 33.6 | 2,521 | 66.4 | |
| No | 85,611 | 95.8 | 20,534 | 24.0 | 65,077 | 76.0 | |
| Diabetes | | | | | | | < 0.0001 |
| Yes | 1,302 | 1.5 | 424 | 32.6 | 878 | 67.4 | |
| No | 88,103 | 98.5 | 21,383 | 24.3 | 66,720 | 75.7 | |
| COVID-19 as secondary diagnosis | | | | | | | 0.9408 |
| Yes | 2,777 | 3.1 | 679 | 24.5 | 2,098 | 75.5 | |
| No | 86,628 | 96.9 | 21,128 | 24.4 | 65,500 | 75.6 | |

Source: Ministry of Health–The SUS Inpatient Care Information System

according to the Charlson and Elixhauser comorbidity indices or patients with diseases such as hypertension, diabetes and obesity, which show a pattern consistent with expectations.

Table 1 also features a variable created to detect any differences in mortality between patients with COVID-19 as the primary diagnosis or as a secondary diagnosis. Hospitalizations with the COVID-19 diagnosis registered in one of the secondary diagnoses corresponded to

**Table 2. Bivariate analyses of variables related to the inpatient healthcare process and in-hospital mortality.** COVID-19 hospitalizations in the Unified Health System in Brazil (N = 89,405). February to June 2020.

| Variables | N | % | In-hospital death | | | | $\chi^2$ (p-value) |
|---|---|---|---|---|---|---|---|
| | | | Yes | | No | | |
| | | | N | % | N | % | |
| Length of stay (days) | | | | | | | < 0.0001 |
| 0–1 | 10,558 | 11.8 | 3,399 | 32.2 | 7,159 | 67.8 | |
| 2–7 | 49,842 | 55.7 | 9,993 | 20.0 | 39,849 | 80.0 | |
| 8–22 | 25,975 | 29.1 | 7,429 | 28.6 | 18,546 | 71.4 | |
| ≥ 23 | 3,030 | 3.4 | 986 | 32.5 | 2,044 | 67.5 | |
| ICU use | | | | | | | < 0.0001 |
| Yes | 20,204 | 22.6 | 11,247 | 55.7 | 8,957 | 44.3 | |
| No | 69,201 | 77.4 | 10,560 | 15.3 | 58,641 | 84.7 | |

Source: Ministry of Health–The SUS Inpatient Care Information System

**Table 3. Bivariate analyses of macro context variables and in-hospital mortality.** COVID-19 hospitalizations in the Unified Health System in Brazil (N = 89,405). February to June 2020.

| Variables | N | % | In-hospital death | | | | $\chi^2$ (Valor de p) |
|---|---|---|---|---|---|---|---|
| | | | Yes | | No | | |
| | | | N | % | N | % | |
| Hospital ownership | | | | | | | < 0.0001 |
| Public–Municipality | 37,539 | 42.0 | 7,544 | 20.1 | 29,995 | 79.9 | |
| Pubic–State | 28,728 | 32.1 | 8,903 | 31.0 | 19,825 | 69.0 | |
| Public–Federal | 958 | 1.1 | 313 | 32.7 | 645 | 67.3 | |
| Private | 4,119 | 4.6 | 862 | 20.9 | 3,257 | 79.1 | |
| Philanthropic | 18,061 | 20.2 | 4,185 | 23.2 | 13,876 | 76.8 | |
| State | | | | | | | < 0.0001 |
| Rondônia | 957 | 1.1 | 127 | 13.3 | 830 | 86.7 | |
| Acre | 64 | 0.1 | 28 | 43.8 | 36 | 56.3 | |
| Amazonas | 4,682 | 5.2 | 1,628 | 34.8 | 3,054 | 65.2 | |
| Roraima | 677 | 0.8 | 243 | 35.9 | 434 | 64.1 | |
| Pará | 3,398 | 3.8 | 647 | 19.0 | 2,751 | 81.0 | |
| Amapá | 237 | 0.3 | 106 | 44.7 | 131 | 55.3 | |
| Tocantins | 265 | 0.3 | 55 | 20.8 | 210 | 79.2 | |
| Maranhão | 5,946 | 6.6 | 1,644 | 27.6 | 4,302 | 72.4 | |
| Piauí | 1,060 | 1.2 | 115 | 10.8 | 945 | 89.2 | |
| Ceará | 6,141 | 6.9 | 1,472 | 24.0 | 4,669 | 76.0 | |
| Rio Grande do Norte | 917 | 1.0 | 238 | 26.0 | 679 | 74.0 | |
| Paraíba | 848 | 0.9 | 166 | 19.6 | 682 | 80.4 | |
| Pernambuco | 8,524 | 9.5 | 2,370 | 27.8 | 6,154 | 72.2 | |
| Alagoas | 804 | 0.9 | 246 | 30.6 | 558 | 69.4 | |
| Sergipe | 271 | 0.3 | 44 | 16.2 | 227 | 83.8 | |
| Bahia | 2,853 | 3.2 | 780 | 27.3 | 2,073 | 72.7 | |
| Minas Gerais | 2,987 | 3.3 | 500 | 16.7 | 2,487 | 83.3 | |
| Espírito Santo | 1,932 | 2.2 | 559 | 28.9 | 1,373 | 71.1 | |
| Rio de Janeiro | 10,589 | 11.8 | 3,401 | 32.1 | 7,188 | 67.9 | |

*(Continued)*

**Table 3.** (Continued)

| Variables | N | % | In-hospital death | | | | | χ² (Valor de p) |
|---|---|---|---|---|---|---|---|---|
| | | | Yes | | No | | | |
| | | | N | % | N | % | | |
| São Paulo | 28,396 | 31.8 | 6,235 | 22.0 | 22,161 | 78.0 | | |
| Paraná | 1,753 | 2.0 | 291 | 16.6 | 1,462 | 83.4 | | |
| Santa Catarina | 988 | 1.1 | 147 | 14.9 | 841 | 85.1 | | |
| Rio Grande do Sul | 2,173 | 2.4 | 325 | 15.0 | 1,848 | 85.0 | | |
| Mato Grosso do Sul | 81 | 0.1 | 7 | 8.6 | 74 | 91.4 | | |
| Mato Grosso | 384 | 0.4 | 76 | 19.8 | 308 | 80.2 | | |
| Goiás | 902 | 1.0 | 124 | 13.7 | 778 | 86.3 | | |
| Distrito Federal | 1,576 | 1.8 | 233 | 14.8 | 1,343 | 85.2 | | |
| Population size | | | | | | | | < 0.0001 |
| ≤ 50.000 hab. | 8,671 | 9.7 | 878 | 10.1 | 7,793 | 89.9 | | |
| 50.001–100.000 hab. | 7,639 | 8.5 | 1,573 | 20.6 | 6,066 | 79.4 | | |
| 100.001–500.000 hab. | 23,442 | 26.2 | 5,785 | 24.7 | 17,657 | 75.3 | | |
| > 500.000 hab. | 49,653 | 55.5 | 13,571 | 27.3 | 36,082 | 72.7 | | |
| Residence/hospitalization | | | | | | | | < 0.0001 |
| Same municipality | 68,069 | 76.1 | 15,565 | 22.9 | 52,504 | 77.1 | | |
| Different municipalities | 21,336 | 23.9 | 6,242 | 29.3 | 15,094 | 70.7 | | |

Source: Ministry of Health–The SUS Inpatient Care Information System

3.1% of the sample, and in this case, the bivariate analyses did not reveal any differences with respect to the occurrence of death.

In Table 2, the percentage of hospitalizations of less than 24 hours or of one day (32.2%), as well as the high mortality of this category (11.8%), are salient statistics. Approximately two thirds of the total hospitalizations lasted up to 7 days (29.1%), between 8 and 22 days (3.4%) and 23 days or more (3.4%). Based on the bivariate analyses, there appears to be a greater concentration of deaths in the extreme categories of LOS, with a lower occurrence of deaths (20.0%) among those with LOS between 2 and 7 days. The proportion of deaths among those who used the ICU was high (55.7%) compared to those who did not (15.3%).

It is worth highlighting the difference between the LOS of those who did not die and those who died during hospitalization. In the first group, the average LOS was 6.7 (± 6.2) days, and the median was 5 days. Among the patients who died, the average LOS was 7.6 (± 7.2) days, and the median was 6 days.

Table 3 shows that most hospitalizations for COVID-19 occurred in municipal public hospitals (42%), followed by state public hospitals (32.1%) and philanthropic hospitals (20.2%). The share of private hospitals contracted by SUS was 4.6%, while that of federal hospitals was 1.1%.

The distribution of hospitalizations by state presents, to a certain extent, a representative picture of the period analyzed, in which some states in the Southeastern, Northeastern and Northern regions were most affected by the epidemic. Just under a third (31.8%) of the hospitalizations analyzed occurred in São Paulo, the largest and wealthiest Brazilian state. São Paulo was followed by Rio de Janeiro (11.8%), Pernambuco (9.5%), Ceará (6.9%), Maranhão (6.7%) and Amazonas (5.2%). Among these states, the occurrence of in-hospital deaths was especially high in Amazonas (34.1%) and Rio de Janeiro (32.1%), with figures also higher than the national average (24.4%) in Pernambuco (27.8%) and Maranhão (27.6%). The Amazonian

states of Acre, Roraima and Amapá corresponded to 0.1%, 0.8% and 0.3%, respectively, of hospitalizations in the country, but they are states with relatively small populations, which stood out for the high proportions of observed in-hospital deaths—43.8%, 35.9% and 44.7%, respectively. The states of Alagoas (30.6%), Espírito Santo (28.9%), Bahia (27.3%) and Rio Grande do Norte (26, 8%) also had in-hospital death percentages higher than the national average.

Table 3 also shows that 81.7% of admissions for COVID-19 in the first four months of the epidemic in the country occurred in municipalities with more than 100 thousand inhabitants, with more deaths observed in these municipalities than in other smaller ones. More than ¾ of hospitalizations were carried out in hospitals located in the same municipality of residence of the patient, with a higher occurrence of deaths when the patient had to travel to receive hospital care.

Table 4 presents the three regression models that explain the occurrence of in-hospital death, considering the progressive inclusion of the "blocks" of variables already mentioned. The first line of the table provides the variance of random intercepts related to hospital units for each model. In general terms, we observe that some patterns of mortality among categories of variables change from the descriptive analyses to the multivariate models, given the control for confounding factors. There is also consistency in the results from adding "blocks" of variables from one model to the next, although, strictly speaking, the 'race/color' mixed race variable loses statistical significance between the second and third models.

Considering model 3, it is possible to observe that the odds of in-hospital death among men were 16.8% higher than among women. The patients' age groups were very important predictors of the likelihood of death. Compared to patients between 18 and 39 years old, patients aged 40 to 49 years were 54.7% more likely to die in the hospital, while for patients aged 90 years or more this increase reached 1,604.9%. Furthermore, blacks had higher odds of death during the hospital stay (OR = 1.14; 95% CI 1.05–1.25), compared to the reference category including whites, yellows, indigenous and individuals without a record of the variable. The adjusted risk of in-hospital death for mixed race people was not statistically significant at the 5% level, but "borderline."

Despite the substantial underreporting of clinical conditions, the behavior of the Charlson and Elixhauser indices was consistent with the hypothesis of higher likelihood of death among patients with comorbidities. The odds of death were 37.2% and 88.1% (model 3) higher among patients with Charlson index scores equal to 1 and ≥2, respectively, compared to those with a score equal to zero, controlling for other variables. For those with Elixhauser comorbidities, the adjusted odds of dying were 41.1% higher than for those without such comorbidities. Presence of hypertension (OR = 0.85; 95% CI 0.73–0.99) and diabetes (OR = 0.75; 95% CI 0.61–0.91) had protective effects, regardless of the inclusion of the other comorbidity measures. Obesity was statistically associated with higher odds of in-hospital death (56.3% higher among obese people, compared to non-obese people). Finally, patients who had COVID-19 as a secondary diagnosis, liable, in some cases, to have acquired the infection in the hospital itself, had 14.9% higher odds of dying in the hospital than those for whom the disease was registered as the main diagnosis.

Some changes were observed in the second regression model when compared to what was observed in the bivariate analyses of LOS and occurrence of death (Table 2). Controlling for other variables, the higher odds of in-hospital death (OR = 3.58; 95% CI 3.35–3.83) for those whose stay was up to 1 day continued to be apparent. Compared to patients with LOS between 8 and 22 days, those with LOS between 2 and 7 days were more likely to die (OR = 1.28; 95% CI 1.22–1.34), while patients with a hospital stay of at least 23 days were less likely to die (OR = 0.66; 95% CI 0.60–0.72). ICU use was a relevant predictor of higher likelihood of in-hospital death (OR = 11.19; 95% CI 10.61–11.81).

**Table 4. Logistic regression models with the factors associated with the variation in in-hospital mortality in COVID-19 hospitalizations in the Unified Health System (N = 89,405).** Brazil, February to June 2020.

| Variable | Model 1 | | | | | Model 2 | | | | | Model 3 | | | | |
|---|---|---|---|---|---|---|---|---|---|---|---|---|---|---|---|
| | Estimate | Standard Error | OR | 95%CI | | Estimate | Standard Error | OR | 95%CI | | Estimate | Standard Error | OR | 95%CI | |
| $\hat{\sigma}^2$ | 1.254 | 0.064 | - | - | - | 1.038 | 0.056 | - | - | - | 0.870 | 0.051 | - | - | - |
| Intercept | -3.113 | 0.049 | - | - | - | -4.035 | 0.054 | - | - | - | -4.839 | 0.102 | - | - | - |
| Sex | | | | | | | | | | | | | | | |
| Male | 0.181 | 0.018 | 1.199 | 1.157 | 1.242 | 0.155 | 0.020 | 1.168 | 1.124 | 1.213 | 0.155 | 0.020 | 1.168 | 1.124 | 1.214 |
| Female | - | - | 1.000 | - | - | - | - | 1.000 | - | - | - | - | 1.000 | - | - |
| Ethnic group | | | | | | | | | | | | | | | |
| Black | 0.173 | 0.042 | 1.189 | 1.094 | 1.291 | 0.149 | 0.046 | 1.160 | 1.061 | 1.269 | 0.136 | 0.046 | 1.145 | 1.047 | 1.253 |
| Mixed race | 0.052 | 0.025 | 1.053 | 1.002 | 1.106 | 0.068 | 0.027 | 1.070 | 1.014 | 1.129 | 0.040 | 0.027 | 1.041 | 0.986 | 1.098 |
| Other | - | - | 1.000 | - | - | - | - | 1.000 | - | - | - | - | 1.000 | - | - |
| Age (years) | | | | | | | | | | | | | | | |
| 18–39 | - | - | 1.000 | - | - | - | - | 1.000 | - | - | - | - | 1.000 | - | - |
| 40–49 | 0.411 | 0.043 | 1.508 | 1.387 | 1.640 | 0.432 | 0.046 | 1.541 | 1.409 | 1.685 | 0.436 | 0.046 | 1.547 | 1.414 | 1.692 |
| 50–59 | 0.841 | 0.039 | 2.318 | 2.147 | 2.502 | 0.854 | 0.042 | 2.349 | 2.164 | 2.550 | 0.859 | 0.042 | 2.360 | 2.174 | 2.562 |
| 60–69 | 1.396 | 0.037 | 4.037 | 3.751 | 4.345 | 1.416 | 0.040 | 4.120 | 3.806 | 4.459 | 1.422 | 0.040 | 4.144 | 3.829 | 4.486 |
| 70–79 | 1.805 | 0.038 | 6.077 | 5.641 | 6.547 | 1.866 | 0.041 | 6.464 | 5.965 | 7.004 | 1.873 | 0.041 | 6.507 | 6.004 | 7.052 |
| 80–89 | 2.233 | 0.041 | 9.325 | 8.597 | 10.115 | 2.380 | 0.045 | 10.802 | 9.895 | 11.792 | 2.391 | 0.045 | 10.923 | 10.004 | 11.927 |
| $\geq 90$ | 2.544 | 0.061 | 12.734 | 11.288 | 14.364 | 2.825 | 0.066 | 16.868 | 14.835 | 19.180 | 2.836 | 0.066 | 17.049 | 14.991 | 19.391 |
| Charlson Index | | | | | | | | | | | | | | | |
| 0 | - | - | 1.000 | - | - | - | - | 1.000 | - | - | - | - | 1.000 | - | - |
| 1 | 0.330 | 0.072 | 1.391 | 1.208 | 1.602 | 0.322 | 0.079 | 1.379 | 1.182 | 1.609 | 0.317 | 0.079 | 1.372 | 1.176 | 1.601 |
| $\geq 2$ | 0.641 | 0.094 | 1.898 | 1.579 | 2.280 | 0.635 | 0.104 | 1.888 | 1.540 | 2.314 | 0.632 | 0.104 | 1.881 | 1.536 | 2.305 |
| Elixhauser comorbidities | | | | | | | | | | | | | | | |
| Yes | 0.278 | 0.073 | 1.321 | 1.145 | 1.523 | 0.359 | 0.080 | 1.431 | 1.224 | 1.674 | 0.345 | 0.080 | 1.411 | 1.207 | 1.651 |
| No | - | - | 1.000 | - | - | - | - | 1.000 | - | - | - | - | 1.000 | - | - |
| Obesity | | | | | | | | | | | | | | | |
| Yes | 0.616 | 0.100 | 1.851 | 1.521 | 2.252 | 0.448 | 0.112 | 1.565 | 1.258 | 1.947 | 0.447 | 0.111 | 1.563 | 1.256 | 1.944 |
| No | - | - | 1.000 | - | - | - | - | 1.000 | - | - | - | - | 1.000 | - | - |
| Hypertension | | | | | | | | | | | | | | | |
| Yes | -0.120 | 0.072 | 0.887 | 0.770 | 1.021 | -0.157 | 0.079 | 0.855 | 0.732 | 0.998 | -0.157 | 0.079 | 0.854 | 0.732 | 0.997 |
| No | - | - | 1.000 | - | - | - | - | 1.000 | - | - | - | - | 1.000 | - | - |
| Diabetes | | | | | | | | | | | | | | | |
| Yes | -0.233 | 0.093 | 0.792 | 0.660 | 0.950 | -0.296 | 0.102 | 0.744 | 0.609 | 0.909 | -0.289 | 0.102 | 0.749 | 0.613 | 0.915 |
| No | - | - | 1.000 | - | - | - | - | 1.000 | - | - | - | - | 1.000 | - | - |
| COVID-19 as secondary diagnosis | | | | | | | | | | | | | | | |
| Yes | 0.153 | 0.058 | 1.165 | 1.040 | 1.305 | 0.149 | 0.063 | 1.161 | 1.026 | 1.314 | 0.139 | 0.063 | 1.149 | 1.015 | 1.301 |
| No | - | - | 1.000 | - | - | - | - | 1.000 | - | - | - | - | 1.000 | - | - |
| Length of stay (days) | | | | | | | | | | | | | | | |
| 0–1 | - | - | - | - | - | 1.263 | 0.034 | 3.537 | 3.308 | 3.780 | 1.276 | 0.034 | 3.582 | 3.350 | 3.829 |
| 2–7 | - | - | - | - | - | 0.238 | 0.023 | 1.269 | 1.214 | 1.327 | 0.244 | 0.023 | 1.276 | 1.221 | 1.335 |
| 8–22 | - | - | - | - | - | - | - | 1.000 | - | - | - | - | 1.000 | - | - |
| $\geq 23$ | - | - | - | - | - | -0.418 | 0.049 | 0.658 | 0.598 | 0.725 | -0.423 | 0.049 | 0.655 | 0.595 | 0.722 |
| ICU use | | | | | | | | | | | | | | | |

*(Continued)*

**Table 4.** (Continued)

| Variable | Model 1 | | | | | Model 2 | | | | | Model 3 | | | | |
|---|---|---|---|---|---|---|---|---|---|---|---|---|---|---|---|
| | Estimate | Standard Error | OR | 95%CI | | Estimate | Standard Error | OR | 95%CI | | Estimate | Standard Error | OR | 95%CI | |
| Yes | - | - | - | - | - | 2.431 | 0.027 | 11.374 | 10.784 | 11.997 | 2.415 | 0.027 | 11.192 | 10.609 | 11.806 |
| No | - | - | - | - | - | - | - | 1.000 | - | - | - | - | 1.000 | - | - |
| Hospital ownership | | | | | | | | | | | | | | | |
| Public—State | - | - | - | - | - | - | - | - | - | - | 0.479 | 0.085 | 1.615 | 1.366 | 1.909 |
| Public—Federal | - | - | - | - | - | - | - | - | - | - | 0.430 | 0.234 | 1.538 | 0.972 | 2.434 |
| Private | - | - | - | - | - | - | - | - | - | - | -0.073 | 0.173 | 0.930 | 0.662 | 1.306 |
| Philanthropic | - | - | - | - | - | - | - | - | - | - | 0.247 | 0.080 | 1.281 | 1.095 | 1.498 |
| Public—Municipality | - | - | - | - | - | - | - | - | - | - | - | - | 1.000 | - | - |
| State | | | | | | | | | | | | | | | |
| Acre | - | - | - | - | - | - | - | - | - | - | 1.690 | 0.581 | 5.418 | 1.736 | 16.909 |
| Amazonas | - | - | - | - | - | - | - | - | - | - | 1.123 | 0.189 | 3.073 | 2.121 | 4.453 |
| Pará | - | - | - | - | - | - | - | - | - | - | 0.666 | 0.156 | 1.947 | 1.434 | 2.644 |
| Amapá | - | - | - | - | - | - | - | - | - | - | 2.346 | 0.536 | 10.441 | 3.649 | 29.871 |
| Maranhão | - | - | - | - | - | - | - | - | - | - | 0.433 | 0.146 | 1.542 | 1.158 | 2.052 |
| Ceará | - | - | - | - | - | - | - | - | - | - | 0.681 | 0.131 | 1.976 | 1.528 | 2.556 |
| Rio Grande do Norte | - | - | - | - | - | - | - | - | - | - | 0.970 | 0.231 | 2.639 | 1.677 | 4.152 |
| Paraíba | - | - | - | - | - | - | - | - | - | - | 0.676 | 0.301 | 1.966 | 1.090 | 3.545 |
| Pernambuco | - | - | - | - | - | - | - | - | - | - | 0.526 | 0.126 | 1.693 | 1.322 | 2.167 |
| Alagoas | - | - | - | - | - | - | - | - | - | - | 0.962 | 0.268 | 2.618 | 1.547 | 4.429 |
| Bahia | - | - | - | - | - | - | - | - | - | - | 0.330 | 0.166 | 1.392 | 1.006 | 1.925 |
| Rio de Janeiro | - | - | - | - | - | - | - | - | - | - | 0.827 | 0.124 | 2.286 | 1.794 | 2.912 |
| São Paulo | - | - | - | - | - | - | - | - | - | - | 0.214 | 0.089 | 1.239 | 1.041 | 1.475 |
| Other | - | - | - | - | - | - | - | - | - | - | - | - | 1.000 | - | - |
| Population size | | | | | | | | | | | | | | | |
| < 100.000 hab. | - | - | - | - | - | - | - | - | - | - | - | - | 1.000 | - | - |
| ≥ 100.000 hab. | - | - | - | - | - | - | - | - | - | - | 0.543 | 0.069 | 1.721 | 1.503 | 1.972 |
| Residence/ hospitalization | | | | | | | | | | | | | | | |
| Same municipality | - | - | - | - | - | - | - | - | - | - | -0.099 | 0.026 | 0.906 | 0.861 | 0.952 |
| Different municipalities | - | - | - | - | - | - | - | - | - | - | - | - | 1.000 | - | - |
| -2 Res Log Pseudo-Likelihood | 447143.5 | | | | | 467927.5 | | | | | 468080.5 | | | | |
| C statistics | 0.6942 | | | | | 0.8017 | | | | | 0.8179 | | | | |

Source: Ministry of Health–The SUS Inpatient Care Information System

Regarding the third model, which included contextual variables related to the organizational aspect of the hospital and its geographic location, the results point to a greater likelihood of in-hospital deaths in state public hospitals and philanthropic hospitals, compared to municipal public hospitals, in addition to a group of Brazilian states where in-hospital mortality due

to COVID-19 was more critical in the period under scrutiny. The hospitalized patients in the states of Amazonas, Rio Grande do Norte, Alagoas, Rio de Janeiro, Ceará, Paraíba, Pará, Pernambuco and Maranhão, were at least 50% more likely to die than in other states in the reference category (predominantly states in the Southern and Midwestern regions), controlling for other variables. The highest odds ratios were, however, observed for Acre (OR = 5.42; 95% CI 1.74–16.91) and Amapá (OR = 10.44; 95% CI 3.65–29.87), which, due to the small number of hospitalizations, had estimates with very broad confidence intervals.

Finally, the odds of death during hospitalization were 72.1% higher in municipalities with at least 100 thousand inhabitants and being admitted to a hospital in the same municipality of residence remained a protective factor for the outcome variable considered.

Based on the statistics related to the goodness of fit of the models, the three blocks of variables in the models constituted explanatory factors for the variation in in-hospital deaths. The attributes of the patients themselves allowed for a reasonable predictive capacity of the model (c = 0.69), which significantly increased with the inclusion of variables related to the care process (c = 0.80). Then, a more modest improvement was observed with the inclusion of the variables related to the hospital's organizational context and geographic area (c = 0.82).

## Discussion

The study provides a comprehensive overview of COVID-19 hospitalizations that occurred in the SUS, including 89,405 hospitalizations, among which 24.4% resulted in death. By focusing on the exploration of explanatory factors for the occurrence of death during hospitalizations due to COVID-19, it contributes with relevant findings to the international debate, confirming knowledge that has been consolidated, raising questions and exposing specificities of the Brazilian context.

Sociodemographic factors and the presence of comorbidities have been identified as associated with COVID-19 hospitalization and death [20, 21]. Similar to what was described in other studies, males, older age group gradually higher, black race/color, Charlson score, presence of Elixhauser comorbidity and obesity presented higher adjusted odds of death [7, 8, 22–24].

This study ratifies the association of a higher risk of COVID-19 in-hospital mortality with being black, as reported in other studies [10, 25, 26]. It is more ambiguous, however, with regard to the risk differentiation of mixed race people, contrasting with the study published by Baqui et al, which even attributed higher risk to mixed race people than to blacks [10]. In Brazil, color/racial differences are correlated with socioeconomic conditions, and blacks and mixed race people are in general more vulnerable than whites. In the specific context of COVID-19, they are still likely to expose themselves more often to the virus [27]. Nevertheless, in the country as a whole, the mixed race color may express a wide spectrum of ethnic mix, which could result in some imprecision and blur effects. It is noteworthy that some studies have looked for explanations for the higher mortality among blacks beyond the socioeconomic aspects, accounting for pathophysiological mechanisms. One relationship of interest is that among being black, COVID-19 and the risk of venous thrombosis [28].

The low level of comorbidity reporting (21,7% of the hospitalizations) in our data is a weakness in the study. It corresponds to just over a quarter of the proportion of patients with at least one chronic disease among those admitted to hospital in a study from New York [20]. It probably reflects some negligence in relation to clinical information in administrative data, but also a negative culture of underreporting, aggravated by the stressful conditions for COVID-19 patient attendance. In spite of the problem, the Charlson and Elixhauser indices, as expected, were shown to have positive associations with the in-hospital mortality risk, and obesity was shown to increase that risk, irrespective of other factors [29]. The protective effects of

hypertension and diabetes in the multivariate models seem paradoxical and inconsistent with some reports in the literature [17, 30], but may also reflect the control for the Charlson and Elixhauser indices. In fact, in a review of the relationship between hypertension and the use of angiotensin-converting enzyme (ACE) inhibitors with COVID-19 outcomes, the authors argue that there is no evidence to support the hypothesis that hypertension or inhibitors of the renin-angiotensin system contribute to unfavorable outcomes in viral infections [31].

The median LOS of 5 days is consistent with data in the United States [17], but differs substantively from data in Lombardy, the Italian region most affected by the pandemic in the first months of 2020, with a median of 28 days of hospitalization [32]. Findings of this study indicate how the characteristics of the patients affect the relationship between LOS and in-hospital mortality. Controlling for confounding variables, the occurrence of shorter hospitalizations is significantly associated with the occurrence of death. Especially, the high risk of death in the first 24 hours of hospitalization, may reflect problems patients had to access timely inpatient care, as well as the quality of care immediately received. Among the hospitalizations that used the ICU, the adjusted odds ratio of death was extreme (OR 11.74), probably reflecting the severity of the cases, but also some synergy with the quality of care. In a meta-analysis that included 24 studies from three continents, a combined mortality rate of 41.6% of patients admitted to the ICU was observed, a value well below the 55.7% found in this study [4].

In the context of the pandemic, the SUS hospital network has been crucial for responding to the demands for acute care that emerged. However, numerous problems related to health service structural conditions and performance have arisen, including the insufficient number of hospital beds and staff to perform specialized care in the ICU [8, 33, 34]. There was, consequently, a broad variation in healthcare effectiveness. In Northern Brazil, at the beginning of the COVID-19 outbreak, states such as Acre, Roraima, Amazonas, Pará and Amapá featured municipalities with exceptionally low or no capacity whatsoever to treat severe cases of the disease [33], which is reflected in the high adjusted odds of death, especially in Amapá, Acre and Amazonas. In Northeastern Brazil, Rio Grande do Norte was the state with the highest odds of in-hospital death, with only three municipalities with minimal capacity to deal with severe cases of the infection at the beginning of the pandemic. Despite the greater availability of hospital beds in Southern and Southeastern Brazil, São Paulo and Rio de Janeiro were hard hit by the pandemic and Rio de Janeiro, in particular, had comparatively very high odds of in-hospital mortality, besides other serious outcomes. There was poor interaction between the state and municipalities severely affected, such as in the capital, the management of the pandemic was chaotic, and the strategy of implementing field hospitals failed partially, with some never completed and others delivered too late.

Our study has limitations, the main issue being the source of information used. The SIH only covers the SUS hospital network, which makes it impossible to carry out a more comprehensive analysis, including healthcare received by those privately insured. It is likely that differences in COVID-19 in-hospital mortality arose reflecting inequities in supply and access to critical resources in specific states of the country [35]. In addition to this, the data flow from providers to the system, and the subsequent consolidation of the information, is slower than desirable to monitor the care provided in a pandemic context that requires swift decisions. Issues regarding the sufficiency and quality of the information recorded should also be stressed, notably the high underreporting of comorbidities and the 'race/color' variable. Furthermore, it was not possible to include cases treated in the emergency wards, and data on the evolution of cases (such as vital signs), and on the care process (professionals involved, use of invasive mechanical ventilation and laboratory tests, including tests for the detection of COVID-19) are absent from this source, which precludes more detailed analysis. Moreover,

the study does not cover deaths that occurred outside hospitals, which constitutes an important statistic to fully grasp the scale of the pandemic morbidity and mortality scenario.

Despite the limitations mentioned, the study has the merit of examining in-hospital mortality with national coverage of the COVID-19 patients who were admitted to hospitals and received care from the SUS, thereby enabling the assessment of the effects of individual and contextual risk factors. Although the source of information, design and statistical modeling limit comparability, the findings broadly corroborate those highlighted by Baqui et al. (2020) regarding regional and racial/ethnic variation in the Brazilian context [10]. In addition, although there is a vast and growing literature on COVID-19, there are still few attempts to address the issue with the strategies we used, focusing on the profile and outcomes of COVID-19 hospitalizations nationwide and painstakingly assess the effects of the groups of variables [10, 17]. In the Brazilian context, the socioeconomic gradient emerges, even with the limits of the data on race/color and geographic location to trace the multiple facets of the inequalities in society [36]. The broad disparities in the performance of the health system among states also becomes apparent. This is related, in part, not only to the structure and prior organization of the services available, but also to the insufficient regional/local capacity to coordinate actions to deal with COVID-19, in the absence of national coordination able to mitigate the major regional differences in an immense and diverse country.

It is of paramount importance to emphasize that this study addresses hospitalizations in the initial months of the pandemic, reflecting the major crisis faced by some capitals, especially in the North, Northeast and Southeast of Brazil, with a high caseload and insufficient healthcare capacity. Covid-19 clinical management, predominantly for severe cases, has subsequently evolved. It is to be expected that the same analyses in subsequent months might provide another overview of how the country has been affected.

With the results provided, we hope to contribute to the improvement of the care delivered and to define strategies to face future developments in the progress of the pandemic until the population has access to an effective vaccine. It is important to remember that the pandemic has evolved dynamically throughout the country.

## Acknowledgments

The authors are grateful for support from PrInt Fiocruz-CAPES Program. CCAP, MM e MCP are recipients of productivity fellowships from the Brazilian National Council of Scientific and Technological Development (CNPq).

## Author Contributions

**Conceptualization:** Carla Lourenço Tavares de Andrade, Claudia Cristina de Aguiar Pereira, Mônica Martins, Sheyla Maria Lemos Lima, Margareth Crisóstomo Portela.

**Data curation:** Carla Lourenço Tavares de Andrade, Claudia Cristina de Aguiar Pereira, Mônica Martins, Margareth Crisóstomo Portela.

**Formal analysis:** Carla Lourenço Tavares de Andrade, Claudia Cristina de Aguiar Pereira, Mônica Martins, Sheyla Maria Lemos Lima, Margareth Crisóstomo Portela.

**Methodology:** Carla Lourenço Tavares de Andrade, Claudia Cristina de Aguiar Pereira, Mônica Martins, Margareth Crisóstomo Portela.

**Project administration:** Margareth Crisóstomo Portela.

**Software:** Carla Lourenço Tavares de Andrade.

**Supervision:** Mônica Martins, Margareth Crisóstomo Portela.

**Validation:** Carla Lourenço Tavares de Andrade, Claudia Cristina de Aguiar Pereira, Margareth Crisóstomo Portela.

**Writing – original draft:** Carla Lourenço Tavares de Andrade, Claudia Cristina de Aguiar Pereira, Mônica Martins, Sheyla Maria Lemos Lima, Margareth Crisóstomo Portela.

**Writing – review & editing:** Carla Lourenço Tavares de Andrade, Claudia Cristina de Aguiar Pereira, Mônica Martins, Sheyla Maria Lemos Lima, Margareth Crisóstomo Portela.

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
