## [Decision Letter · Decision Letter 0]

30 Sep 2020

PONE-D-20-27175

COVID-19 hospitalizations in Brazil’s Unified Health System (SUS)

PLOS ONE

Dear Dr. Portela,

Thank you for submitting your manuscript to PLOS ONE. After careful consideration, we feel that it has merit but does not fully meet PLOS ONE’s publication criteria as it currently stands. Therefore, we invite you to submit a revised version of the manuscript that addresses the points raised during the review process.

Two reviewers have assessed the manuscritpt and provide discordant decisons. The main reasons for reject the paper is some concerns about study validity. Please, the new version should include specific information about study validity to be the manuscript considered to publication.    

We look forward to receiving your revised manuscript.

Kind regards,

Bruno Pereira Nunes, Ph.D.

Academic Editor

PLOS ONE

Journal Requirements:

3. If you are reporting a retrospective study of medical records or archived samples, please ensure that you have discussed whether all data were fully anonymized before you accessed them.

4. Please note that according to our submission guidelines (http://journals.plos.org/plosone/s/submission-guidelines), outmoded terms and potentially stigmatizing labels should be changed to more current, acceptable terminology. For example: “Pardo” should be changed to “mixed race” (as appropriate).

Reviewers' comments:

Reviewer's Responses to Questions

**Comments to the Author**

1. Is the manuscript technically sound, and do the data support the conclusions?

Reviewer #1: No

Reviewer #2: Yes

2. Has the statistical analysis been performed appropriately and rigorously? 

Reviewer #1: No

Reviewer #2: Yes

3. Have the authors made all data underlying the findings in their manuscript fully available?

Reviewer #1: Yes

Reviewer #2: Yes

4. Is the manuscript presented in an intelligible fashion and written in standard English?

Reviewer #1: No

Reviewer #2: Yes

5. Review Comments to the Author

Reviewer #1: It's a large study covering Unified Health System in the COVID-19 pandemic. But I feel it's lack of novelty and failed to provide new information regarding risk factors of in-hospital mortality in patients with COVID-19. You mentioned that 78.3% of hospitalizations did not have any secondary diagnoses, which made me concern that there might be misdocumentation affecting the validity of the data. Also, it might be better to consider length of stay and ICU use as outcomes.

Reviewer #2: The theme chosen for the study was of great relevance. The Title and absctract were adequate and consistent.

The introduction presented fundamental characteristics in its composition. As an example, I point out the description of the unique health system, peculiar to Brazil and necessary for the understanding of the study. I advise that the text between lines 77 to 99 could be summarized.

The objectives were clear. As a suggestion, I would include the last sentence after the objectives (lines 116 to 118) as a strong point of the work and not at the end of this paragraph.

The methods have been well described. I advise you to describe the Charlson and Elixhauser indices in this topic and not in the results.

The results reported consistent and very important topics.

We advise you to keep the discussion more concise, valuing the most important results.

6. PLOS authors have the option to publish the peer review history of their article (what does this mean?). If published, this will include your full peer review and any attached files.

Reviewer #1: No

Reviewer #2: No

---

## [Author Response · Author response to Decision Letter 0]

5 Nov 2020

EDITOR’S AND REVIEWERS’ COMMENTS AND AUTHOR’S ANSWERS

EDITOR

Comment: Thank you for submitting your manuscript to PLOS ONE. After careful consideration, we feel that it has merit but does not fully meet PLOS ONE’s publication criteria as it currently stands. Therefore, we invite you to submit a revised version of the manuscript that addresses the points raised during the review process.

Two reviewers have assessed the manuscript and provide discordant decisions. The main reasons for reject the paper is some concerns about study validity. Please, the new version should include specific information about study validity to be the manuscript considered to publication. 

Answer: The concerns with the validity of the study aroused especially from the underreporting of comorbidities, but there were also the indications, by one of the two reviewers, of “lack of novelty and failed to provide new information regarding risk factors of in-hospital mortality”. 

Our first point is that administrative data may be really important economic and prompt sources in observational studies in the Health Services Research field. The SUS Hospital Information System (SIH), as other administrative data worldwide, have been broadly employed in scientific studies, despite the inherent limitations. The SIH is the main source of information on hospital production in Brazil. It includes all hospitalizations in the Brazilian public system, responsible for covering 75% of the Brazilian population in the whole country. The anonymized SIH database is freely accessible in DATASUS website, and, to the best of our knowledge, our study is the first to use them to picture COVID-19 pandemic in Brazil. Even having some flow problems and not being available in real time, the data is mostly available in relatively short time. In the course of a pandemic in which evidence still needs to be accumulated, and Brazil has unquestionable importance as, at this point, the third country in number of cases and second in number of deaths, the role it can play is not negligible. At the same time, we are honest and explicitly recognize in the text the weaknesses of the data, arguing that they are supplanted by the contributions the study brings out. 

Despite the efforts made in the last years to expand the number of fields for secondary diagnoses from one to nine in the SIH, they are not fulfilled. We truly have reasons to believe, based in previous studies and the experience as researchers, that there is a bad culture of clinical data underreporting in the country. If the study were based on the medical records in the hospitals, we would still have a lot of missing data. Hospitals with research activities often keep separate data for patients included in studies. The alternative of only be confident on primary data, however, would be unpractical and would neglect the capacity of secondary data provide useful information. 

In our data, 78.3% of hospitalizations did not have a secondary diagnosis. It is likely that the COVID-19 scenario had even contributed for worse underreporting of comorbidities. In fact, data misrecording is expected in urgency contexts as the pandemic. Despite the problem, the death predictive capacity of model 1, the one which accounts for the clinical predictors of inpatient mortality, was 0.69, borderline to the range that is considered adequate (0.70-0.80). 

We developed the study from the perspective of health services researchers and not clinicians. The level of hospital mortality is seen as an outcome that result not only from clinical aspects, but also the quality of healthcare. It was not our goal to necessarily provide new information on clinical risk factors of in-hospital mortality, but the available evidence was employed and tested in the modeling process. The focus was to compare COVID-19 hospital mortality adjusted by severity, expressing variations in quality of care – effectiveness and access inequalities – between hospital ownership and geographic areas.

Even considering the papers available, more evidence on care delivered in different countries seems important yet. The high volume of inpatients analyzed, severity adjusted strategies, focus on quality of care, and comparing geographical areas inequalities are the strong points. 

Comment: Please ensure that your manuscript meets PLOS ONE's style requirements, including those for file naming. The PLOS ONE style templates can be found at https://journals.plos.org/plosone/s/file?id=wjVg/PLOSOne_formatting_sample_main_body.pdf and https://journals.plos.org/plosone/s/file?id=ba62/PLOSOne_formatting_sample_title_authors_affiliations.pdf

Answer: This was done.

Comment: We suggest you thoroughly copyedit your manuscript for language usage, spelling, and grammar. If you do not know anyone who can help you do this, you may wish to consider employing a professional scientific editing service.

Answer: This was done.

The English reviewer was Derrick Guy Phillips (Tel.: 55-21-99182-0989) – Union-registered translator since 1978 and Sworn Public Translator certified by the Board of Trade in Rio de Janeiro since March 2010. / M.A. – Master of Arts Degree ‘summa cum laude’ in Modern Languages & Philosophy (French, Spanish, Portuguese & Philosophy) from the University of St Andrews. Scotland (September 1969 through June 1973). / Post-Graduate Diploma in “International Marketing for Language Graduates” from the University of Central London, England (September 1977 through June 1978) – Group Award and Special Mention. / Translator and Director of Feedback Traduções Ltda. (Translation Agency), since 1978. 

Comment: If you are reporting a retrospective study of medical records or archived samples, please ensure that you have discussed whether all data were fully anonymized before you accessed them.

Answer: The data obtained were already anonymized.

Comment: Please note that according to our submission guidelines (http://journals.plos.org/plosone/s/submission-guidelines), outmoded terms and potentially stigmatizing labels should be changed to more current, acceptable terminology. For example: “Pardo” should be changed to “mixed race” (as appropriate).

Answer: This was done.

Comment: In your Data Availability statement, you have not specified where the minimal data set underlying the results described in your manuscript can be found. PLOS defines a study's minimal data set as the underlying data used to reach the conclusions drawn in the manuscript and any additional data required to replicate the reported study findings in their entirety. All PLOS journals require that the minimal data set be made fully available. For more information about our data policy, please see http://journals.plos.org/plosone/s/data-availability.

Answer: OK. The data is open access, as described in the manuscript. We are making the data used in the study available. Anyway, we can make the data we employed in the study available following instructions of the journal.

REVIEWERS

1. Is the manuscript technically sound, and do the data support the conclusions?

Reviewer #1: No

Reviewer #2: Yes

Answer: As we have already indicated in this letter and in the manuscript, we acknowledge that the study has some limitations. However, we are also convinced that it provides a good overview of COVID-19 hospitalizations in the Brazilian Unified Health System in the first four months of the pandemic in Brazil and highlights the huge variation in in-hospital mortality, associated with social and healthcare quality inequities throughout the country. We maintain that our research is sound and the data support all conclusions provided. The study is replicable.

2. Has the statistical analysis been performed appropriately and rigorously? 

Reviewer #1: No

Reviewer #2: Yes

Answer: We believe that we used the best statistical approach available for the research question at hand. We were interested in analyzing factors associated with in-hospital mortality due to Covid-19 and applied variables commonly used in the health service research literature to address this type of question. Furthermore, we were careful to use appropriate modelling to account for the violation of the assumption of independence among observations in the same hospital.

3. Have the authors made all data underlying the findings in their manuscript fully available?

Reviewer #1: Yes

Reviewer #2: Yes

Answer: OK.

4. Is the manuscript presented in an intelligible fashion and written in standard English?

Reviewer #1: No

Reviewer #2: Yes

Answer: We are providing a new version of the manuscript revised by an English native speaker and professional translator.

5. Review Comments to the Author

Reviewer #1: It's a large study covering Unified Health System in the COVID-19 pandemic. But I feel it's lack of novelty and failed to provide new information regarding risk factors of in-hospital mortality in patients with COVID-19. You mentioned that 78.3% of hospitalizations did not have any secondary diagnoses, which made me concern that there might be misdocumentation affecting the validity of the data. Also, it might be better to consider length of stay and ICU use as outcomes.

Answer: To the best of our knowledge, this is the first study using data from the SUS Hospital Information System, an administrative database which covers, nationally, all hospitalizations that take place in the SUS, that provides healthcare to 75% of the Brazilian population. We were able to analyze 89,405 hospitalizations across the country in the first four months of the COVID-19 pandemic in Brazil. Considering the importance of Brazil in the global COVID-19 pandemic scene, we would probably be making a bigger mistake if we neglected information which can be extracted from the database, and would have no better source, especially if swiftness and economic criteria were applied. 

Despite the underreporting of comorbidities, the strong gradient observed for age in all models, which has even increased along the inclusion of the blocks of variables, seems to indicate age as a more reliable indicator to predict in-hospital death in the context. 

Such analyses are needed to shine a light on in-hospital mortality and associated factors. Although much has been discussed in the media and in the academic output about comorbidities such as hypertension, obesity and diabetes, we found some novel results such as: the surprisingly higher chance of death when the length of stay was short (less than 24 hours), probably reflecting access barriers; higher odds ratio adjusted by severity in some areas, indicating effectiveness variability; the in-hospital mortality risk related to transference to another city for healthcare; specific differences among hospitals with different kinds of ownership.

Also, it is true that the important race gradient (higher mortality among blacks, followed by mixed race people) had already been shown in other studies, but it is likely to reflect marked social inequities that need to be exposed and addressed by policy makers. Accumulating evidence and understanding how COVID-19 outcomes are affected by other factors beyond clinical factors is the overriding goal.

With regard to the comment about the inclusion of length of stay (LOS) and ICU use as explanatory variables, we justify our choice underlining our health services researchers’ perspective. Both variables used are healthcare processing variables. ICU use ends up reflecting, to some extent, clinical severity, beyond the comorbidity data itself. LOS, in turn, seems to reflect healthcare access and quality. Our study design and analyses were not oriented towards detecting causality relations in which cause precedes effect. By including both variables, we sought to identify simply how the risk of in-hospital mortality varies vis-à-vis distinct standards of hospital stay duration and ICU use or not. 

A data quality gap is to be expected in all sources of information, even medical records. The emergency room, ICU and wards were operating in crisis mode and under dramatic conditions. Based on the abovementioned points, we believe that all possible information should be obtained and examined to draw a more complete picture of the care provided, in various contexts and countries.

Reviewer #2: The theme chosen for the study was of great relevance. The Title and abstract were adequate and consistent.

The introduction presented fundamental characteristics in its composition. As an example, I point out the description of the unique health system, peculiar to Brazil and necessary for the understanding of the study. I advise that the text between lines 77 to 99 could be summarized.

The objectives were clear. As a suggestion, I would include the last sentence after the objectives (lines 116 to 118) as a strong point of the work and not at the end of this paragraph.

The methods have been well described. I advise you to describe the Charlson and Elixhauser indices in this topic and not in the results. The results reported are consistent and very important topics. We advise you to keep the discussion more concise, valuing the most important results.

Answer: Thank you for your careful review of our manuscript and comments about the adequacy of our title, abstract, objectives and results. We think that we attended all your recommendations.

We summarized the text between lines 77 and 99 of the previous version to nine lines (73-81) in the new version.

We also accepted your suggestion transferring the last sentence after the objectives to the end of the Discussion section, as a strong point of the work. 

In fact, the descriptions and references of the Charlson and Elixhauser indices were already in the Methods section. We have highlighted them in the file with track changes, and added a few more explanations to make the text clearer. No explanations about them were left in the results. 

We have worked on the Discussion and made it shorter by emphasizing the results that we considered most important.

---

## [Editor Report · Decision Letter 1]

17 Nov 2020

COVID-19 hospitalizations in Brazil’s Unified Health System (SUS)

PONE-D-20-27175R1

Dear Dr. Portela,

We’re pleased to inform you that your manuscript has been judged scientifically suitable for publication and will be formally accepted for publication once it meets all outstanding technical requirements.

Kind regards,

Bruno Pereira Nunes, Ph.D.

Academic Editor

PLOS ONE
---

## [Editor Report · Acceptance letter]

23 Nov 2020

PONE-D-20-27175R1 

COVID-19 hospitalizations in Brazil’s Unified Health System (SUS) 

Dear Dr. Portela:

I'm pleased to inform you that your manuscript has been deemed suitable for publication in PLOS ONE. Congratulations! Your manuscript is now with our production department. 

Kind regards, 

on behalf of

Dr. Bruno Pereira Nunes 

Academic Editor

PLOS ONE